# Improving Outcomes in the Advanced Gastrointestinal Stromal Tumors: The Role of the Multidisciplinary Team Discussion Intervention

**DOI:** 10.3390/jpm13030417

**Published:** 2023-02-26

**Authors:** Pan Ran, Hui Zhou, Jinjin Li, Tao Tan, Hao Yang, Juan Li, Jun Zhang

**Affiliations:** 1Department of Gastrointestinal Surgery, The First Affiliated Hospital of Chongqing Medical University, Chongqing 400016, China; 2Department of Internal Medicine, Chongqing Key Laboratory of Translation Research for Cancer Metastasis and Individualized Treatment, Chongqing University Cancer Hospital, Chongqing 400030, China; 3Department of Pharmacy, The First Affiliated Hospital of Chongqing Medical University, Chongqing 400016, China

**Keywords:** gastrointestinal stromal tumor, multidisciplinary team, overall survival, performance status, prognosis

## Abstract

Objectives: There is disagreement over the prognostic value of multidisciplinary team (MDT) discussion for advanced gastrointestinal stromal tumors (GISTs). This study examined how an MDT affected patients with advanced GISTs in terms of their overall survival (OS) and whether it may enhance their performance status (PS). Methods: A retrospective data analysis was conducted on patients with advanced GISTs between 2000 and 2022. Depending on whether they had received the MDT discussion intervention, the patients were split into two groups. The OS between the two groups was compared using the Kaplan–Meier method. A multivariate Cox regression analysis was used to analyze the prognostic variables for advanced GIST. Fisher’s test was used to investigate the relationship between an MDT and PS. Results: There were 122 patients with an MDT and 117 patients without an MDT in this study. In comparison to the non-MDT group, the MDT group showed a higher survival rate (5-year OS, 42.62% vs. 28.21%, *p* < 0.05). MDT was an independent prognostic factor for OS in univariate and multivariate Cox regression analyses (*p* < 0.05). Fisher’s test revealed that there were variations in PS between the two groups (*p* < 0.05). Conclusions: The effectiveness of an MDT in the treatment of advanced GIST was examined for the first time in this study. MDT discussion intervention is an effective measure for improving the outcomes of patients with advanced GISTs.

## 1. Introduction

Gastrointestinal stromal tumors (GISTs) are the most common mesenchymal tumors of the gastrointestinal tract and have malignant potential [1,2]. Although the exact incidence of GIST is unknown, it has been increasing [3]. Importantly, many patients with GISTs had unresectable or metastatic tumors at first diagnosis [4,5]. Even in patients with primary localized GISTs after completing surgical resection, the 5-year recurrence rate is 70.5% [6]. As described above, metastatic and recurrent diseases are commonly observed in patients with GISTs [7]. While patients with primary localized GISTs may receive curative treatment by surgical resection and postoperative adjuvant therapy with imatinib [8], patients with advanced GISTs are not as lucky. Radical surgical resection of the tumor may not be feasible for patients with advanced GISTs [9]. Luckily, however, the clinical application of various targeted drugs has transformed the acute threat of deadly cancer into a manageable chronic condition [10]. Therefore, attention should be paid to prolonging the survival time and improving the performance status (PS) of patients with GISTs. 

Most clinicians make treatment decisions based on clinical experience and existing treatment guidelines, but due to complex strategies in the treatment of patients with GISTs, accurate diagnosis and treatment are difficult to achieve [11,12]. To address the above issue, the multidisciplinary team (MDT) was established. The MDT discussion intervention consists of a group of medical experts and specialists from a range of disciplines, working together to provide comprehensive care to patients, which could provide personalized and targeted treatment tailored to each person’s expectations, condition, and situation [11,12]. MDT discussion intervention is thus increasingly recommended by medical treatment units to improve the prognosis of patients with advanced GISTs [13,14,15]. However, due to the lack of relevant clinical cohort studies to prove that MDTs can prolong patients’ survival times, demonstrating the value of MDT discussion intervention in patients with advanced GISTs may be important. Therefore, the purpose of this study was to analyze the effect of MDT on the overall survival (OS) of patients with advanced GISTs and to evaluate whether MDT could improve the PS of patients with GISTs.

## 2. Methods

### 2.1. Basic Characteristics of Patients

Clinical data on all patients with advanced GISTs were obtained from the First Affiliated Hospital of Chongqing Medical University (CMU; Chongqing, China) from 1 January 2000, to 1 April 2022. The follow-up continued until 1 July 2022. The criteria for inclusion were as follows: (1) GIST confirmed by biopsy or postoperative pathology, and (2) age over 18 years. The criteria for exclusion were as follows: (1) incomplete clinical data or loss to follow-up; (2) follow-up time < 3 months; (3) patients with GISTs who had undergone complete tumor resection and had no recurrence of the tumor at the end of follow-up; and (4) a history or presence of other malignancies. The patient screening process used in this study is illustrated in Figure 1. Patients who met the criteria for inclusion and exclusion were classified into MDT and non-MDT groups according to whether they had experienced MDT discussion intervention or not.

We created a database called “Weinichangzai”, which included relevant clinical data for each patient with GIST. Data collected included patients’ mutation genotype, primary tumor site, gender, age, Ki-67 labeling index (Li), and expression of DOG-1, CD117, and CD34. All patients with GISTs in this study were in an advanced state, and the tumor metastasis site, medication history, operation history, etc., were not uniform, which indicated that the treatment of patients with advanced GISTs is not static. Thus, these factors were not included in the analysis due to the fact that their treatment histories cannot be simply presented. This study was approved by the Institutional Review Board of the First Affiliated Hospital of CMU (approval number: 2022-K364) with a waiver for written informed consent, owing to its observational and retrospective design. This study was conducted under the principles of the Declaration of Helsinki.

### 2.2. Follow-Up and Operational Definition

The GIST patients were followed up every 3–6 months in the GIST specialist clinic. From 1 July 2022, to 1 August 2022, alive patients with advanced GISTs were followed up in the GIST specialist clinic or by phone, and their PS was assessed using the Eastern Cooperative Oncology Group (ECOG) PS score [16]. 

The prognosis endpoints were OS and ECOG PS score; OS was defined as the time from the patient’s diagnosis of advanced GIST to the death resulting from any cause, so it did not include recurrence-free survival time. Patient age refers to the patient’s age at the time of the diagnosis of advanced GIST. The value of Ki-67 Li was obtained from the pathology report of a resected specimen when available or from that of a biopsy specimen if no surgery had been conducted. The expression of DOG-1, CD117, and CD34 was also obtained from the pathology report of a resected specimen when available or from that of a biopsy specimen if no surgery had been conducted.

### 2.3. MDT for GIST

The MDT discussion was moderated by the chief physician of the gastrointestinal surgery department; the hepatobiliary surgeon, gastroenterologist, oncologist, pathologist, radiologist, oncologist, pharmacist, and clinical molecular medical testing center physician participated in the discussion. The MDT discussion intervention was routinely held once a month in a designated demonstration classroom. The consultation, initial diagnosis, and follow-up of patients with GISTs were mainly the responsibility of gastrointestinal surgeons. The doctors gathered the imaging data and the patient’s medical history before the MDT discussion. Generally, the discussion was as follows: review the patient history, identify the diagnosis and genotype, manage the severe adverse reactions caused by targeted therapy, evaluate the tumor progression, discuss the timing and protocol of surgery, etc. Subsequently, according to the disease situation of each patient with GIST, the best treatment plan was proposed under collective wisdom to achieve the purpose of the individualized treatment of patients with GISTs; it was also conducive to the standardization of diagnosis and treatment. It was important that the implementation of the MDT discussion intervention be made with the consent of the patient with GIST to participate in the discussion.

Additionally, after the discussion, there was a special secretary to record and summarize each MDT discussion in detail. A strategy for treating each patient with GIST was developed following the MDT discussion. Some examples of case management using the MDT discussion intervention are presented in Appendix A.

### 2.4. Statistical Analysis

The baseline characteristics of the study population were analyzed using descriptive statistics. The Kolmogorov–Smirnov test was used to check for the normal distribution of continuous variables. We calculated variable distributions using means ±SD for continuous variables that satisfied a normal distribution, the median (interquartile range) for continuous variables that satisfied a non-normal distribution, and frequencies (percentages) for categorical variables. Comparisons of categorical variables were made using Pearson’s chi-square test or Fisher’s exact test. The difference in continuous variables between groups was assessed by a Student’s t-test (normal distribution) or Mann–Whitney test (non-normal distribution). The OS of patients with advanced GISTs was analyzed using the Kaplan–Meier survival curve (log-rank test). A univariate Cox analysis was performed for each prognostic variable, and variables with *p* < 0.2 in this analysis were included in the multivariate Cox analysis. Differences were considered significant at a two-tailed *p*-value, as indicated by *p* < 0.05. All statistical analyses were performed using SPSS version 27.0 (IBM Corp, Armonk, NY, USA).

## 3. Results

### 3.1. Baseline Data

After applying the inclusion and exclusion criteria, 239 patients with advanced GISTs were eligible for this study (Figure 1). There were 122 patients (51%) in the MDT group and 117 patients (49%) in the non-MDT group. The average follow-up time and OS time (excluding recurrence-free survival time) were 70.02 and 53.13 months, respectively. The longest follow-up time was 268.83 months. In the MDT group, 77 males and 45 females were aged 55.38 ± 11.65 years, while in the non-MDT group, 74 men and 43 females were aged 57.44 ± 12.98 years. The primary tumor sites included 61 gastric (25.52%), 104 small intestinal (43.51%), 15 colorectal (6.28%), and 59 non-gastrointestinal (24.69%) tumors. The positivity rates of DOG-1, CD117, and CD34, were 89.12%, 97.91%, and 81.59%, respectively. The baseline clinical characteristics of the two groups are shown in Table 1. The mutation genotype differed at baseline between the MDT and non-MDT groups (*p* < 0.05). Age, Ki-67 Li, gender, primary tumor site, expression of DOG-1, expression of CD117, and expression of CD34 did not show a significant difference between the two groups. Table 1 shows the difference in baseline clinical characteristics between the MDT group and the non-MDT group. Column percentages may not sum to 100% due to rounding error.

### 3.2. Impact of MDT on OS in Patients with GISTs

The Kaplan analysis showed a statistically significant difference in mortality risk between the MDT and non-MDT groups in patients with advanced GISTs (log-rank, *p* = 0.014). Based on the result, the median OS was 48.13 months for the MDT group and 37.83 months for the non-MDT group. The OS rate at 1 year, 3 years, and 5 years for the non-MDT group was 84.62%, 50.43%, and 28.21%, respectively, while those of the MDT group were 95.90%, 68.85%, and 42.62%, respectively (Figure 2). In the non-MDT group, only six patients died before an MDT was established.

A multivariate Cox regression analysis was used to eliminate confounding-factor interference because the baseline characteristics were unbalanced between the MDT and non-MDT groups. We analyzed nine prognostic factors (age, Ki-67 Li, gender, primary tumor site, mutation genotype, expression of DOG-1, expression of CD117, expression of CD34, and group) using a univariate Cox regression analysis. Four factors (age, primary tumor site, expression of DOG-1, and group, *p* < 0.2) from the univariate Cox analysis were included in the multivariate Cox analysis to perform the statistical analysis. In both univariate and multivariate Cox analyses, MDT discussion intervention was a protective factor that decreased the mortality of patients with advanced GISTs (HR = 0.622, *p* < 0.05, Table 2). It is worth emphasizing that, though age is significant in multivariate Cox analysis (*p* < 0.05), it is not significant in univariate Cox analysis (*p* = 0.06). From the analysis results of multivariate Cox analysis, it can be seen that older age is a detrimental factor that increases mortality in advanced GIST.

### 3.3. Choice of Treatment Methods after MDT Discussion

In 78 MDT meetings, 122 advanced GIST patients (77 males and 45 females) were discussed 233 times. Figure 3 shows a pie chart detailing the percentage of choice of treatment methods for advanced GIST after the MDT discussion intervention, which was based on the number of discussions. In summary, the choice of different treatment methods for advanced GIST after MDT discussion intervention can be summarized as follows: maintenance treatment (41%), replacement of targeted drugs (20%), palliative surgery (16%), perfect inspection (9%), adjusting the dosage of targeted drugs (7%), intensive follow-up (4%), and visits to specialist departments (3%).

### 3.4. MDT and PS

The percentage of GIST patients with ECOG performance status scores of 0, 1, 2, 3, 4, and 5 in the MDT group was 4.10%, 51.64%, 4.92%, 4.10%, 0.82%, and 34.43%, respectively. The percentage of GIST patients with ECOG performance status scores of 0, 1, 2, 3, 4, and 5 in the non-MDT group was 5.98%, 34.19%, 17.09%, 2.56%, 2.56%, and 37.61%, respectively. The MDT group had a significantly better PS than the non-MDT group, and the difference in the PS between the two groups was statistically significant, regardless of whether the patients who died were excluded (*p* < 0.05, Figure 4). Column percentages may not sum to 100% due to rounding error.

## 4. Discussion

The most likely estimate of GIST incidence is >12 cases per 10^6^ persons per year [17]. Additionally, based on the available data, GISTs with KIT mutations have an incidence of nearly 8 cases per 10^6^ individuals per year, and GISTs with PDGFRA mutations have an incidence <3 cases per 10^6^ individuals per year [17]. Surgical resection is the only potentially curative treatment for primary localized GISTs [18], but in patients with advanced GISTs, radical surgical resection of the tumor may not be possible [9]. However, fortunately, tyrosine kinase inhibitors (TKIs), which are the first targeted drugs of choice for patients with advanced GISTs, have significantly improved patient outcomes [19]. According to different indications, a number of targeted drugs have been identified, including imatinib, sunitinib, regorafenib, and other targeted drugs [17]. The clinical application of a variety of targeted drugs has transformed the acute threat of deadly cancer into a manageable chronic disease [10].

Therefore, attention should be paid to prolonging the survival time and improving the PS of patients with GISTs. The therapeutic management of GIST not only includes surgical but also systemic treatments, which is the driving force for multidisciplinary collaboration [17]. Thus, GIST cannot just be diagnosed and treated through a single discipline. In order to effectively manage any disease, the most appropriate treatment should be administered in a timely manner, and treatment modalities should be periodically corrected as needed [11,20]. Therefore, on the basis of this, the concept of the MDT was proposed, gathering a variety of medical experts from multiple disciplines to ensure the best diagnosis and treatment options for patients with advanced GISTs, and to reduce the rate of misdiagnosis, missed diagnosis, and mistreatment, so that patients receive standard and accurate treatment [11].

Previous studies have also suggested that an MDT may be essential for the successful treatment of GIST, and the multidisciplinary disease management of patients may prolong their survival and reduce delays in treatment and referral [11,21,22,23]. Bareck et al. reported that the treatment of GISTs needed interdisciplinary management, taking into account the complex strategies for the management of patients with GISTs. Endoscopy, histopathology, radiology, surgery, and oncology were topics in the treatment of patients with GISTs. Particularly in cases of advanced GISTs, our multidisciplinary knowledge is needed [24]. However, due to the lack of relevant cohort studies to support this conclusion, the clinical benefits of MDT discussion interventions for patients with advanced GISTs have not been well proven. This study was intended to fill this gap. 

Therefore, a retrospective method was used to compare the baseline clinical characteristics and prognosis of patients with advanced GISTs who did and did not receive the MDT discussion intervention in this study. What is interesting in this study is that a difference was found between the MDT and non-MDT groups in baseline clinical characteristics, such as mutation genotype (*p* < 0.05, Table 1). Based on this, it may be easier for MDTs to discuss medical records with patients with KIT exon 9-mutated GISTs. The reason for this phenomenon may be complicated, but the prognosis of patients with KIT exon 9-mutated GISTs is worse than that of patients with other GISTs [25,26], and KIT exon 9-mutated GIST patients are also the only mutation genotype deriving a significant benefit in progression-free survival from the higher imatinib dose (800 mg/day) [26], which may explain the difference between the MDT and non-MDT groups. As the baseline clinical characteristics were unbalanced between the MDT and non-MDT groups, multivariable Cox analyses were performed to control for potential confounders. After the multivariate Cox analysis, MDT discussion intervention remained an independent predictor of OS, which was deemed to be an independent protective factor, reducing the mortality rate by 0.378 times compared to the absence of MDT discussion intervention. We also found that OS was influenced by patient age, with some studies also supporting this finding [27,28].

Obvious achievements have also been made in advanced esophageal cancer, advanced gastric cancer, and advanced colorectal cancer using MDT discussion intervention [29,30,31]. The results of the above studies further validate our conclusions and show that MDT discussion intervention can also be effectively applied to improve outcomes in other complex advanced tumor cases. Similar conclusions are expressed in a review article, in which the benefits of multidisciplinary disease management of patients with GISTs include reduction in recurrent disease, optimization of timing of surgery and organ preservation, prolongation of patient survival, and improved response to targeted therapies, suggesting that an MDT of physicians is critical to the successful treatment of GISTs [11].

Du et al. and Basta et al. found that MDT discussion interventions could often modify the treatment plan to allow for a more holistic treatment, which may significantly improve outcomes for patients with gastrointestinal malignancies [32,33]. For the theories stated above, we have shown a pie chart (Figure 3) detailing the percentage of choices of different treatment methods for patients with advanced GISTs after the MDT discussion intervention. From Figure 3, it can be seen that the percentages of maintenance treatment and treatment of change (including replacement of targeted drugs, palliative operations, perfect inspection, adjusting the dosage of targeted drugs, intensive follow-up, and visits to specialist departments) were 41% and 59%, respectively. The results of some similar studies have shown that after MDT, 13–29% of patients undergo an adjustment to their treatment plan, including their drug treatment plan and surgical treatment plan. The primary factors affecting treatment plan selection in these cases were the re-evaluation of imaging findings and tumor status [34,35,36]. The above results confirmed the role of MDT discussion interventions in providing an individual-based treatment tailored to each individual’s condition, family circumstances, and expectations [11,12].

As for non-MDT patients, they participated in regular follow-up visits at the GIST specialist clinic, and their treatment was often determined by the outpatient physician. Due to their lack of participation in MDT discussions, outpatient physicians may have faced certain risks in the process of diagnosis and treatment for patients with advanced GISTs, whose conditions were complex. For example, as patients with advanced GISTs often receive TKIs for an extended period of time, it is important to maintain and even improve PS, especially as adverse effects can impair medication adherence in the long term, despite the survival advantage improved by the targeted drugs [17]. Figure 4 verifies the above hypothesis. It can be seen from Figure 4 that MDT discussion intervention could aid in maintaining and even improving the PS during tumor treatment, owing to the usual timely adjustment of treatment methods in MDT patients after MDT discussion intervention, which is lacking in non-MDT patients.

This study provides up-to-date insights into multidisciplinary-disease management for patients with GISTs. Additionally, as far as we are aware, the present cohort study is the first to explore the role of an MDT in the management of the diagnosis and treatment of advanced GIST. Multidisciplinary discussions about the impact of interventions on the efficiency of the treatment process are a major concern for many physicians, who are concerned that multiple MDT discussions might delay treatment for patients [30]. However, the results of the paper illustrate that MDT discussion intervention is an effective measure to improve outcomes, including PS and the survival of patients with advanced GISTs. Finally, we recommend that all patients with advanced GISTs undergo MDT discussion intervention to achieve the goal of individual-based treatment and improve prognosis.

However, this study had several limitations. First, it was a single-center study; due to this, it is prudent to extend our results to other centers. Secondly, the study had a retrospective design, which meant that selection bias was unavoidable. For example, because an MDT has different criteria for discussing interventions, it tends to prioritize patients with complex conditions, while patients with more clearly defined conditions tend to be treated without an MDT. Finally, all patients with GISTs in this study were in an advanced state, and the tumor metastasis site, medication history, operation history, etc., were not uniform, which indicated that the treatment of patients with advanced GISTs is not static and, therefore, their treatment history cannot be simply presented. Notwithstanding these limitations, our findings are important as they confirm that MDT intervention improved patient outcomes, including PS and survival.

## 5. Conclusions

In conclusion, we found that the MDT discussion intervention improved outcomes in our study, including PS and survival rate, in patients with advanced GISTs. Disease management guided by MDT is a relatively new concept introduced into clinical practice, particularly in the field of oncology, some of which was changed into a manageable chronic condition. We suggest that all patients with advanced GISTs undergo MDT discussion intervention to achieve the goal of individual-based treatment.

## Figures and Tables

**Figure 1 jpm-13-00417-f001:**
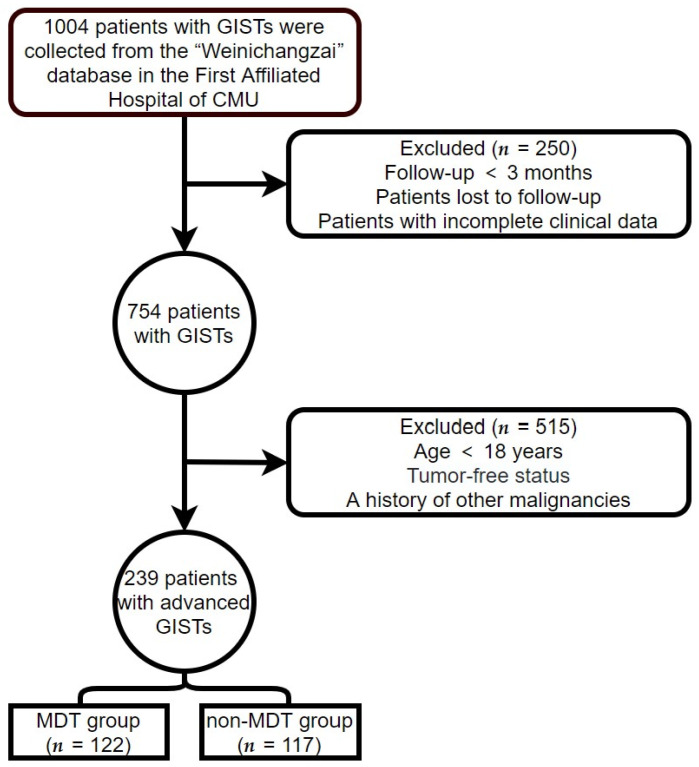
Patient screening process.

**Figure 2 jpm-13-00417-f002:**
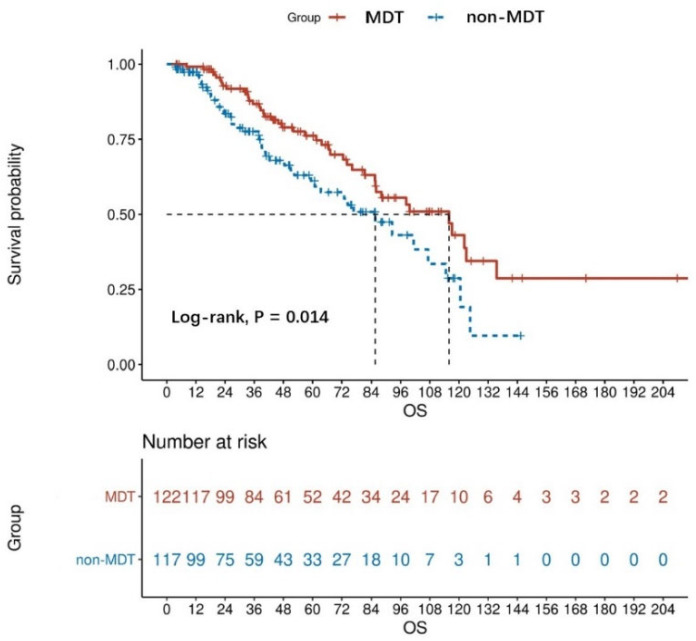
OS curves of MDT (*n* = 122) and non-MDT (*n* = 117) patients with advanced GISTs, the result was obtained from the univariate analysis.

**Figure 3 jpm-13-00417-f003:**
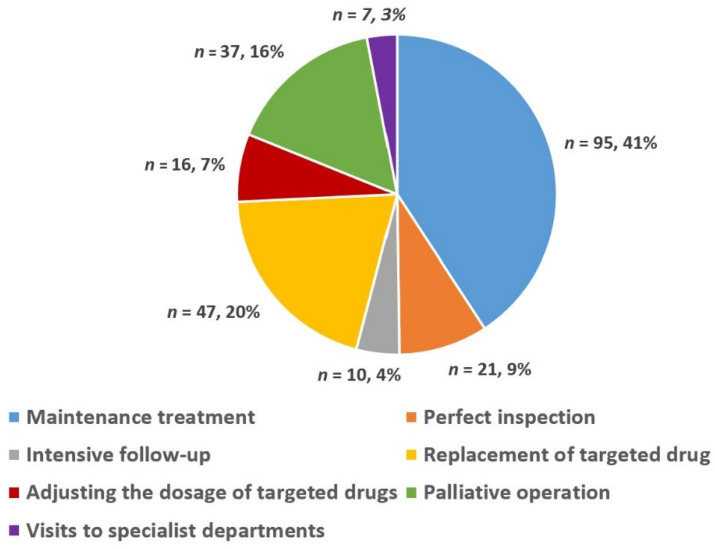
Choice of different treatment methods for advanced GIST after MDT discussion.

**Figure 4 jpm-13-00417-f004:**
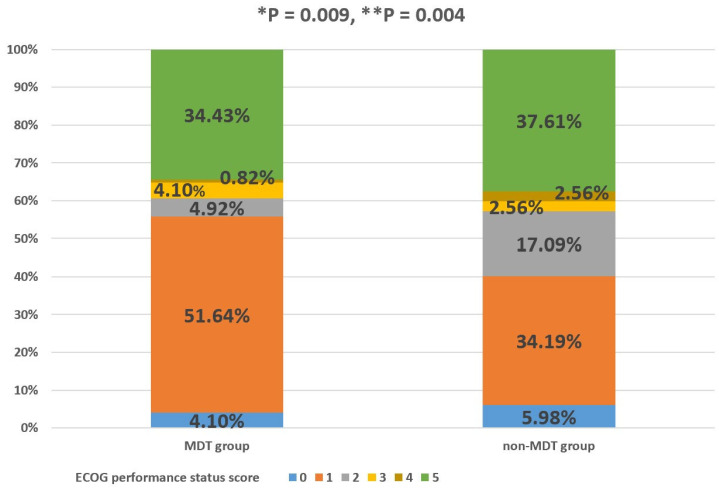
Percentage of MDT and non-MDT groups in each ECOG PS score category. “*” with respect to patients with advanced GISTs who died were also included; “**” with respect to GIST patients who died were excluded. *p*-value was determined using Fisher´s exact test.

**Table 1 jpm-13-00417-t001:** Baseline clinical characteristics of patients with GISTs who received and did not receive MDT discussion intervention.

Characteristics	MDT (*n* = 122)	Non-MDT (*n* = 117)	*p*
Age (year)	55.38 ± 11.65	57.44 ± 12.98	0.198 ^a^
Ki-67 Li (%)	10.00 (10.00)	10.00 (10.00)	0.289 ^b^
Gender			0.983 ^c^
Male	77 (63.11%)	74 (63.25%)	
Female	45 (36.89%)	43 (36.75%)	
Primary tumor site			0.894 ^c^
Gastric	30 (24.59%)	31 (26.50%)	
Small intestine	56 (45.90%)	48 (41.03%)	
Colorectum/rectal	7 (5.74%)	8 (6.84%)	
Other	29 (23.77%)	30 (25.64%)	
Mutation genotype			<0.001 ^d^
KIT exon 11	60 (49.18%)	65 (55.56%)	
KIT exon 9	32 (26.23%)	9 (7.69%)	
Wild type	8 (6.56%)	6 (5.13%)	
Other	5 (4.10%)	1 (0.85%)	
Unknown	17 (13.93%)	36 (30.77%)	
DOG-1			0.762 ^c^
Positive	108 (88.52%)	105 (89.74%)	
Negative	14 (11.48%)	12 (10.26%)	
CD117			0.205 ^d^
Positive	121 (99.18%)	113 (96.58%)	
Negative	1 (0.82%)	4 (3.42%)	
CD34			0.607 ^c^
Positive	98 (80.33%)	97 (82.91%)	
Negative	24 (19.67%)	20 (17.09%)	

Note: ^a^ independent sample t-test; ^b^ Mann–Whitney U-test; ^c^ Chi-square test; ^d^ Fisher’s test; We calculated all *p*-values as two-tailed. Abbreviation: MDT, multidisciplinary team; Li, labeling index; DOG-1, gastrointestinal stromal tumor protein 1; CD117, cluster of differentiation 117; CD34, cluster of differentiation 34.

**Table 2 jpm-13-00417-t002:** Univariate and multivariate Cox analyses of prognostic factors for OS in patients with advanced GISTs.

Characteristic	OS			
	Univariate		Multivariate	
	HR (95% CI)	*p*	HR (95% CI)	*p*
Age (year)	1.017 (0.999–1.036)	0.060	1.019 (1.000–1.038)	0.045
Ki-67 Li (%)	1.006 (0.987–1.025)	0.571	-	
Gender		0.403	-	
Male	1.0 (ref)			
Female	1.203 (0.780–1.856)			
Primary tumor site		0.133		0.136
Gastric	1.0 (ref)		1.0 (ref)	
Small intestine	1.199 (0.722–1.991)		1.207 (0.709–2.057)	
Colorectum/rectal	0.153 (0.021–1.132)		0.148 (0.020–1.099)	
Other	1.441 (0.801–2.594)		1.414 (0.778–2.568)	
Mutation genotype		0.349	-	
KIT exon 11	1.0 (ref)			
KIT exon 9	1.624 (0.918–2.873)			
Wild type	0.503 (0.121–2.090)			
Other	1.448 (0.347–6.050)			
Unknown	1.006 (0.607–1.667)			
DOG-1		0.173		0.208
Negative	1.0 (ref)		1.0 (ref)	
Positive	1.562 (0.822–2.967)		1.521 (0.792–2.920)	
CD117		0.529	-	
Negative	1.0 (ref)			
Positive	0.690 (0.217–2.191)			
CD34		0.815	-	
Negative	1.0 (ref)			
Positive	0.941 (0.565–1.567)			
Group		0.015		0.031
Non-MDT	1.0 (ref)		1.0 (ref)	
MDT	0.588 (0.383–0.902)		0.622 (0.404–0.957)	

Note: “-” indicates no data. Abbreviation: OS, overall survival; HR, hazard ratio; CI, confidence interval; MDT, multidisciplinary team; Li, labeling index; DOG-1, gastrointestinal stromal tumors protein 1; CD117, cluster of differentiation 117; CD34, cluster of differentiation 34.

## Data Availability

The datasets used and analyzed during the current study are available from the corresponding author (zjun@cqmu.edu.cn) on reasonable request.

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
