# Peer review of "Improving Outcomes in the Advanced Gastrointestinal Stromal Tumors: The Role of the Multidisciplinary Team Discussion Intervention"

_jpm, 2023, doi:10.3390/jpm13030417_

Round 1

Reviewer 1 Report

No notes, I hope this kind of articles can underline the importance of MDTs

the abstract could use minor editing so it isn't too similar to this one https://www.mdpi.com/1718-7729/29/2/102/htm

Author Response

Dear Reviewer #2:

    Thanks for your letter and for the reviewer’s comments concerning our manuscript entitled “Improving Outcomes in Advanced Gastrointestinal Stromal Tumors: The Role of Multidisciplinary Team Discussion Intervention” (ID: 2221510). Those comments are all valuable and very helpful for revising and improving our paper, as well as the important guiding significance to our studies. We have studied comments carefully and have made correction which we would like to meet with approval. Revised portion are marked in red in the “Excerpt from revised text”. The main corrections in the paper and the responds to the reviewer’s comments are as flowing: 

Comment: The abstract could use minor editing so it isn't too similar to this one https://www.mdpi.com/1718-7729/29/2/102/htm.

Response: Thank for your minor suggestion. The precedent version of the “abstract” (Line 13-28) has been replaced.

Excerpt from revised text:

Objectives: There is disagreement over the prognostic value of multidisciplinary team (MDT) discussion for advanced gastrointestinal stromal tumors (GISTs). This study examined how an MDT affected patients with advanced GISTs in terms of their overall survival (OS) and whether it may enhance their performance status (PS). Methods: A retrospective data analysis was conducted on patients with advanced GISTs between 2000 and 2022. Depending on whether they had received the MDT discussion, the patients were split into two groups. The OS between the two groups was compared using the Kaplan–Meier method. A multivariate Cox regression analysis was used to analyze the prognostic variables for advanced GIST. Fisher’s test was used to investigate the relationship between an MDT and PS. Results: There were 122 patients with an MDT and 117 patients without an MDT in this study. In comparison to the non-MDT group, the MDT group showed a higher survival rate (5-year OS, 42.62% vs. 28.21%, P < 0.05). MDT was an independent prognostic factor for OS in univariate and multivariate Cox regression analyses (P < 0.05). Fisher’s test revealed that there were variations in PS between the two groups (P < 0.05). Conclusion: The effectiveness of an MDT in the treatment of advanced GIST was examined for the first time in this study. MDT discussion intervention is an effective measure for improving the outcomes of patients with advanced GISTs.

The language of our paper has been handed over to a professional language polishing company (MDPI) for processing.

We tried our best to improve the manuscript and made some changes in the manuscript. These changes will not influence the content and framework of the paper.

We appreciate for Reviewers’ warm work earnestly, and hope that the correction will meet with approval.

Reviewer 2 Report

The authors investigated overall survival (OS) and performance status (PS) of advanced GIST patients. They divided the patients into 2 groups, multidisciplinary (MDT) team and non-MDT team. They concluded patients treated by MDT team showed better OS and PS than non-MDT team.

(Major problem)

・How were the patients divided into MDT and non-MDT group? Why weren’t patients in non-MDT group treated by MDT? I’m afraid that patients in non-MDT group were treated in the old period. If so, not only MDT approach but also improvement of chemotherapy may affect OS.

・I had difficulty to understand the inclusion and exclusion criteria. Did all patients in this study have metastasis or recurrence? As to postoperative cases, was disease-free time included in OS?

(Minor problem)

・In Page 3, “OS follow-up time was up to 12698.66 months” must be wrong.

・Treatment in MDT group should be shown, and the authors should discuss the difference of treatment between MDT and non-MDT group.

Author Response

Dear Reviewer #1:

Thanks for your letter and for the reviewer’s comments concerning our manuscript entitled “Improving Outcomes in Advanced Gastrointestinal Stromal Tumors: The Role of Multidisciplinary Team Discussion Intervention” (ID: 2221510). Those comments are all valuable and very helpful for revising and improving our paper, as well as the important guiding significance to our studies. We have studied comments carefully and have made correction which we would like to meet with approval. Revised portion are marked in red in the “Excerpt from revised text”. The main corrections in the paper and the responds to the reviewer’s comments are as flowing: 

No.1

Comment:The authors investigated overall survival (OS) and performance status (PS) of advanced GIST patients. They divided the patients into 2 groups, multidisciplinary (MDT) team and non-MDT team. They concluded patients treated by MDT team showed better OS and PS than non-MDT team.

Response: We sincerely thank the reviewer for careful reading. As suggested by the reviewer, we have corrected the “general health condition” into “performance status (PS)”.

No.2

Comment:

1. How were the patients divided into MDT and non-MDT groups?

2. Why weren’t patients in the non-MDT group treated by MDT?

3. I’m afraid that patients in the non-MDT group were treated in the old period. If so, not only the MDT approach but also the improvement of chemotherapy may affect OS.

Response:

1. Patients who met the criteria for inclusion and exclusion were classified into MDT and non-MDT groups according to whether they had experienced MDT discussion intervention or not. (Line 62-64)

2. The main purposes of our MDT are as follows: A. Clarify the diagnosis and genotype of GIST, and select the targeted drugs with the best possible efficacy. B. Patients with severe adverse reactions after targeted therapy can be treated in a timely manner. C. To evaluate the progression of the tumor and identify the nature of the tumor outside the primary site. D. Evaluate the appropriate surgical timing and protocol.

Thus, some patients are not treated with MDT for the following reasons: a. Our center has informed all patients that our center has a professional MDT discussion for collaborative diagnosis and treatment, and the implementation of MDT requires the consent of patients, but some patients refused the diagnosis and treatment of MDT. For example, partial patients with severe diseases refuse to participate in MDT discussions and only have regular outpatient follow-up visits. b. Some patients' conditions are not complicated and do not have the above condition (A-D). At the same time, because the implementation of MDT requires the collaborative work of multiple disciplines, and the number of cases discussed at a monthly meeting is limited, not all patients were invited for MDT treatment. c. Only 6 patients in the non-MDT group died before the MDT was established.

  1. In the non-MDT group, only 6 patients were treated in the old period and died before MDT was established. Although our follow-up period was from January 1, 2000 to April 1, 2022, many patients had long periods of recurrence-free survival that were not included in the study, as can be seen from Line 120-121.

Excerpt from revised text:

  1.  

Patients who met the criteria for inclusion and exclusion were classified into MDT and non-MDT groups according to whether they had experienced MDT discussion intervention or not. (Line 62-64)

  1.  

Generally, the discussion is as follows: review the patient history, identify the diagnosis and genotype, manage the severe adverse reactions caused by targeted therapy, evaluate the tumor progression, and discuss the timing and protocol of surgery. Subsequently, according to the disease situation of each patient with GIST, the best treatment plan is proposed under collective wisdom, to achieve the purpose of the individualized treatment of patients with GISTs; it is also conducive to the standardization of diagnosis and treatment. It is important that the implementation of MDT discussion is made with the consent of the patient with GIST to participate in the discussion. (Line 92-100)

3.

In the non-MDT group, only six patients died before an MDT was established. (Line 140-141)

The average follow-up time and OS time (excluding recurrence-free survival time) were 70.02 and 53.13 months, respectively. (Line 120-121)

No.3

Comment:

I had difficulty to understand the inclusion and exclusion criteria. Did all patients in this study have metastasis or recurrence? As to postoperative cases, was disease-free time included in OS?

Response:

We thank the reviewer for the very rigorous comments. In fact, patients with GISTs who had undergone complete tumor resection and had no recurrence of the tumor at the end of follow-up, are already excluded. Therefore, the study population included in this paper are all advanced patients (metastasis or recurrence), so OS does not include disease-free time. To be more clear and in accordance with the reviewer's concerns, we have added a brief description in Line 80-83

Excerpt from revised text:

The prognosis endpoints were OS and ECOG PS score; OS was defined as the time from the patient's diagnosis of advanced GIST to the death resulting from any cause, so it did not include recurrence-free survival time. Patient age referred to the age at the time of diagnosis of advanced GIST.

No.4

Comment:

In Page 3, “OS follow-up time was up to 12698.66 months” must be wrong.

Response:

Thanks for your careful checks. We are sorry for our carelessness. Based on your comments, we have changed the statement in the "Methods" section (Line 120-122).

Excerpt from revised text:

The average follow-up time and OS time (excluding recurrence-free survival time) were 70.02 and 53.13 months, respectively. The longest follow-up time was 268.83 months. (Line 120-122)

No.5

Comment:

  1. Treatment in MDT group should be shown.
  2. The authors should discuss the difference of treatment between MDT and non-MDT group.

Response:

1. The treatment of non-MDT patients is not static. They also participated in regular follow-up visits at the GIST specialist outpatient, and often have changes in the treatment style while in the outpatient or inpatient department, so their treatment history cannot be simply presented. The same way, not all the changes in the treatment methods of patients in the MDT group are shown in Figure 3. They may also have changes in the treatment methods when they participate in the outpatient follow-up of GIST specialist outpatient. Figure 3 only shows the recommended treatment methods during the MDT discussion. So, we have added a brief description in Line 250-253.

2. Thanks for the reviewer’s pertinent suggestion. To be more clear and in accordance with the reviewer concerns, we have added a brief description in Line 232-239

Excerpt from revised text:

1.

Finally, all patients with GISTs in this study were in an advanced state, and the tumor metastasis site, medication history, operation history, etc., were not uniform, which indicated that the treatment of patients with advanced GISTs is not static and, therefore, their treatment history cannot be simply presented.

2.

As for non-MDT patients, they participated in regular follow-up visits at the GIST specialist clinic, and their treatment was often determined by the outpatient physician. Due to the lack of participation in MDT discussions, outpatient physicians may have faced certain risks in the process of diagnosis and treatment for patients with advanced GISTs, whose conditions were complex. Figure 4 verifies the above hypothesis. It can be seen from Figure 4 that MDT discussion intervention could aid in maintaining and even improving the PS during tumor treatment, owing to the usual timely adjustment of treatment methods in MDT patients after MDT discussion, which is lacking in non-MDT patients.

The language of our paper has been handed over to a professional language polishing company (MDPI) for processing.

We tried our best to improve the manuscript and made some changes in the manuscript. These changes will not influence the content and framework of the paper.

We appreciate for Reviewers’ warm work earnestly and hope that the correction will meet with approval.

Round 2

Reviewer 2 Report

Thank you for replying to my comments and revising manuscript. Since the manuscript has been revised properly, I think this manuscript is worth publishing.